# Phenol-Enriched Virgin Olive Oil Promotes Macrophage-Specific Reverse Cholesterol Transport In Vivo

**DOI:** 10.3390/biomedicines8080266

**Published:** 2020-08-03

**Authors:** Lídia Cedó, Sara Fernández-Castillejo, Laura Rubió, Jari Metso, David Santos, Daniel Muñoz-Aguayo, Andrea Rivas-Urbina, Mireia Tondo, Karen Alejandra Méndez-Lara, Marta Farràs, Matti Jauhiainen, Maria-José Motilva, Montserrat Fitó, Francisco Blanco-Vaca, Rosa Solà, Joan Carles Escolà-Gil

**Affiliations:** 1Institut d’Investigacions Biomèdiques IIB Sant Pau, 08041 Barcelona, Spain; daymer11@hotmail.com (D.S.); arivas@santpau.cat (A.R.-U.); mtondo@santpau.cat (M.T.); kmendez1101@gmail.com (K.A.M.-L.); mfarras@santpau.cat (M.F.); fblancova@santpau.cat (F.B.-V.); 2CIBER de Diabetes y Enfermedades Metabólicas Asociadas (CIBERDEM), 28029 Madrid, Spain; 3Surgery Department-Functional Nutrition, Oxidation, and CVD Research Group (NFOC-Salut), Faculty of Medicine and Health Sciences-Medicine, Universitat Rovira i Virgili, 43201 Reus, Spain; sara.fernandez@urv.cat (S.F.-C.); laurarubio@tecal.udl.cat (L.R.); rosa.sola@urv.cat (R.S.); 4Fundació EURECAT—Centre Tecnològic de Nutrició i Salut, 43204 Reus, Spain; 5Food Technology Department, Universitat de Lleida-Agrotecnio Center, 25198 Lleida, Spain; 6Minerva Foundation Institute for Medical Research and National Institute for Health and Welfare, Genomics and Biomarkers Unit, 00290 Helsinki, Finland; jari.metso@thl.fi (J.M.); matti.jauhiainen@thl.fi (M.J.); 7IMIM Hospital del Mar Medical Research Institute, Grup de Risc Cardiovascular i Nutrició, 08003 Barcelona, Spain; DMunoz@imim.es (D.M.-A.); mfito@imim.es (M.F.); 8CIBER of Physiopathology of Obesity and Nutrition CIBEROBN, Grup de Risc Cardiovascular i Nutrició, 28029 Madrid, Spain; 9Departament de Bioquímica i Biologia Molecular, Universitat Autònoma de Barcelona, 08041 Barcelona, Spain; 10Instituto de Ciencias de la Vid y del Vino-ICVV (CSIC-Universidad de La Rioja-Gobierno de La Rioja), Finca “La Grajera”, 26007 Logroño, La Rioja, Spain; motilva@icvv.es; 11Hospital Universitari Sant Joan de Reus HUSJR, NFOC-Salut, 43204 Reus, Spain

**Keywords:** HDL, reverse cholesterol transport, virgin olive oil, phenolic compounds, mice VOHF project

## Abstract

The intake of olive oil (OO) enriched with phenolic compounds (PCs) promotes ex vivo HDL-mediated macrophage cholesterol efflux in humans. We aimed to determine the effects of PC-enriched virgin OO on reverse cholesterol transport (RevCT) from macrophages to feces in vivo. Female C57BL/6 mice were given intragastric doses of refined OO (ROO) and a functional unrefined virgin OO enriched with its own PC (FVOO) for 14 days. Our experiments included two independent groups of mice that received intragastric doses of the phenolic extract (PE) used to prepare the FVOO and the vehicle solution (saline), as control, for 14 days. FVOO intake led to a significant increase in serum HDL cholesterol and its ability to induce macrophage cholesterol efflux in vitro when compared with ROO group. This was concomitant with the enhanced macrophage-derived [^3^H]cholesterol transport to feces in vivo. PE intake *per se* also increased HDL cholesterol levels and significantly promoted in vivo macrophage-to-feces RevCT rate when compared with saline group. PE upregulated the expression of the main macrophage transporter involved in macrophage cholesterol efflux, the *ATP binding cassette*
*a1*. Our data provide direct evidence of the crucial role of OO PCs in the induction of macrophage-specific RevCT in vivo.

## 1. Introduction

The Mediterranean diet, and specifically olive oil (OO) consumption, have been widely demonstrated to provide benefits in terms of cardiovascular disease prevention [1,2]. Such benefits have been mainly attributed to OO nutritional composition; OO has a high proportion of unsaturated fatty acids, mainly monounsaturated oleic acid (55–83%), and a minor unsaponifiable fraction (1%) [3,4] classified into two types: hydrophobic compounds (e.g., tocopherols, pigments) and soluble compounds, which include phenolic compounds (PC) [5]. PCs are comprised of simple phenols (e.g., hydroxytyrosol, tyrosol), secoiridoids (e.g., oleuropein), and polyphenols (e.g., lignans and flavonols) [5]. Evidence indicates that these minor components could provide a major contribution to the cardioprotective benefits of OO consumption [5]. In the “Effect of Olive Oil on Oxidative Damage in a European Population (EUROLIVE)” study, PC-rich OO intake (22 g/day of oil with 366 mg/kg of PC content) in healthy male volunteers increased high-density lipoprotein cholesterol (HDL-C) levels [6] and promoted HDL-mediated cholesterol efflux capacity from macrophages, the first step in reverse cholesterol transport (RevCT) [7]. In 2011, the European Food Safety Authority claimed that “5 mg of hydroxytyrosol and its derivatives (e.g., oleuropein complex and tyrosol) in OO should be consumed daily in order to protect lipids from oxidative damage” [8]. Subsequently, the European Commission (Commission Regulation (EU) No 432/2012; 16 May 2012) established that “in order to bear the claim, information shall be given to the consumer that the beneficial effect is obtained with a daily intake of 20 g of olive oil” [9]. However, the PCs concentration is too low in much commercial virgin OO (VOO) to reach the daily dose of hydroxytyrosol (5 mg) within the context of a balanced diet [8]. Therefore, the intake of OO enriched with its own PC could be of interest to increase the dose of these beneficial compounds without increasing fat intake [10]. A randomized, controlled trial was designed to investigate whether functional VOO enriched with PCs could increase HDL quantity and quality in hypercholesterolemic subjects. In the “Virgin Olive Oil and HDL Functionality” (VOHF) project, hypercholesterolemic subjects ingested VOO (80 mg PC/kg of oil) or functional phenol-enriched VOO (500 mg PC/kg of oil) with either their own PCs (FVOO) or with PCs from OO and additional ones from thyme (FVOOT). Although enriched VOOs did not change the HDL-C levels, the ability of HDL to induce macrophage cholesterol efflux and HDL antioxidant and anti-inflammatory properties were enhanced (reviewed in [11]).

The ability of HDL to promote cholesterol efflux from macrophage foam cells is considered the main HDL atheroprotective function. However, macrophage cholesterol efflux is only the first step of the whole RevCT pathway, followed by the transfer of cholesterol to the liver, from where it is partly eliminated via bile into the intestine and ultimately to feces [12]. The modulation of this atheroprotective pathway has been extensively studied in genetically-engineered mice with the ultimate aim of translating the emerging knowledge to humans [13]. Furthermore, the in vivo macrophage RevCT rate is closely associated with atherosclerosis susceptibility [14]. The effects of phenol-enriched OO on in vivo whole-body macrophage-specific RevCT have not been previously evaluated. In the present study, we determined the potential of a phenol-enriched VOO (thus unrefined as required by the virgin OO EU classification) and its own PC to promote macrophage-to-feces RevCT in vivo.

## 2. Experimental Section

### 2.1. Mice and Treatment

All animal procedures were conducted in accordance with published regulations and reviewed and approved by the Institutional Animal Care Committee of the Institut de Recerca of the Hospital de la Santa Creu i Sant Pau (ref. 9013, approved in April 2016). Eight to twelve-week-old female C57BL/6 wild-type mice were purchased from Jackson Laboratories (Bar Harbor, ME, US; #000664) and kept in a temperature-controlled (22 °C) room with a 12-hour (h) light/dark cycle, while food (regular chow diet A04; Scientific Animal Food & Engineering, Augy, France) and water were provided *ad libitum*. The mice were divided into four groups. Each mouse of the first and second groups was administered daily intragastric doses (150 µL) of (i) refined OO (ROO) and (ii) FVOO (500 mg PCs/kg of oil), respectively. In parallel, the mice of the third and fourth groups received daily intragastric doses (150 µL) of (iii) the phenol extract (PE) used in FVOO preparation and dissolved in a saline solution, or (iv) saline (the vehicle solution was used as PE control), respectively, for 14 days. A similar dose of hydroxytyrosol and its derivatives was administered to the mice in the FVOO and PE groups (Appendix A). The procedure for obtaining the PE and preparing the FVOO was previously described [15]. Briefly, the PE was obtained from a freeze-dried olive cake rich in hydroxytyrosol, the main olive oil PC, and its derivatives. The FVOO, with a total phenolic content of 500 mg/kg of oil, was prepared by the addition of the PE to ROO. The total phenolic content of the FVOO was measured using the Folin-Ciocalteu method [16]. The phenolic profiles of the ROO, FVOO, and PE were analyzed by high-performance liquid chromatography coupled with tandem mass spectrometry (HPLC/MS/MS) using the method previously described [10]. The fatty acid composition of the ROO and FVOO is detailed in Appendix A. Mouse body weight and food intake were monitored in all experimental groups.

### 2.2. Lipid and Apolipoprotein Analyses

Serum was obtained by centrifuging blood for 10 min at 10,000× *g* and total cholesterol was determined enzymatically using commercial kits adapted for a COBAS 6000 autoanalyzer (Roche Diagnostics, Rotkreuz, Switzerland). HDL-C levels were measured in serum obtained after the precipitation of apolipoprotein (APO) B-containing lipoprotein particles with 0.44 mmol/L phosphotungstic acid (Merck, Darmstadt, Germany) and 20 mmol/L magnesium chloride (Sigma-Aldrich, St Louis, MO, USA). The mouse APOA1 assay was quantified with an ELISA kit of 96-well plates coated with a polyclonal rabbit antibody against mouse APOA1, as previously reported [17]. The amount of preß-HDL was quantified with two-dimensional crossed immunoelectrophoresis in individual serum samples incubated for 6 h at 37 °C in the presence of a lecithin–cholesterol acyltransferase (LCAT) inhibitor (1 mmol/L iodoacetate) [17].

### 2.3. Enzyme Activities

Phospholipid transfer protein (PLTP) activity was measured with a radiometric assay using [^14^C]phosphatidylcholine liposomes as donors and HDL_3_ as an acceptor in the presence of the individual serum samples [18]. LCAT activity was measured via a radiometric method using radiolabeled reconstituted APOA1-discoidal HDL particles as substrates and measuring the amount of radiolabeled cholesteryl-ester generated, as described previously [19].

### 2.4. In Vivo Macrophage-to-Feces RevCT

Mouse J774A.1 macrophages (ATCC^®^ TIB67™, Manassas, VA, USA) were cultured in 75-cm^2^ tissue culture flasks (5 × 10^6^ cells/flask) in a RPMI-supplemented medium [2 mmol/L L-glutamine (Pan Biotech, Aidenbach, Germany), 10% fetal bovine serum (FBS; Pan Biotech), and 100 U/mL penicillin/streptomycin (Dominique Dutscher, Brumath, France)]. The cells were then incubated for 48 h in the presence of 5 µCi/mL of [1α,2α(n)-^3^H]cholesterol, 100 µg/mL of acetyl-low-density lipoprotein (acLDL), and 10% human lipoprotein-depleted serum (LPDS). These cells were washed, equilibrated with a medium containing 0.2% bovine serum albumin (BSA; Sigma-Aldrich, Madrid, Spain), detached via scraping, resuspended in PBS, and pooled before being intraperitoneally injected into the mice at the end of the treatment (average of 1.5 × 10^6^ macrophages containing 9 × 10^5^ cpm per mouse; cell viability was 80% measured by trypan blue staining) [20]. The mice were then individually housed in metabolic cages and their stools were collected over the following 48 h. At that point, the mice were euthanized and exsanguinated via cardiac puncture, and their livers were removed. Serum was obtained by centrifuging blood for 10 min at 10,000× *g* and radioactivity was determined at 4, 24, and 48 h via liquid scintillation counting. Liver and fecal lipids were extracted with isopropyl alcohol-hexane (2:3, *v*/*v*), evaporated, and [^3^H]cholesterol radioactivity was measured via liquid scintillation counting [20]. The [^3^H]tracer detected in the fecal bile acids (BAs) was determined for the remaining aqueous portion of the fecal material extracts. The amount of [^3^H]tracer was expressed as a fraction of the injected dose. This method has been used to study the role of different therapies and pathways relevant for RevCT and HDL-mediated atheroprotection [13].

### 2.5. Intestinal Cholesterol Absorption

The efficiency of intestinal cholesterol absorption was determined via the fecal dual isotope ratio method using [^3^H]sitostanol as a nonabsorbable recovery standard in the independent groups of mice given the ROO, FVOO, PE, or saline, as described above. Forty eight hours before the end of treatment, the mice received a stomach bolus of 100 μL ROO containing 1 μCi of [4-^14^C]cholesterol (Perkin Elmer, Waltham, MA, USA) and 2 μCi of [5,6-^3^H]sitostanol (American Radiolabeled Chemicals, St. Louis, MO, USA) as internal control. The mice were then individually housed in metabolic cages, and their stools were collected over the next 48 h (during these 48 h, the mice received the different treatments). At that point, the mice were euthanized and exsanguinated via cardiac puncture. Fecal lipid fraction was extracted with isopropyl alcohol-hexane (2:3, *v*/*v*). The lipid layer was then collected and evaporated. [^3^H]- and [^14^C]-radioactivity were measured via liquid scintillation counting. The fractional cholesterol absorption was calculated according to the formula: ([^14^C]/[^3^H] (in dosing solution) − [^14^C]/[^3^H] (in sample))/[^14^C]/[^3^H] (in dosing solution) × 100.

### 2.6. Ex Vivo Cholesterol Efflux Capacity

The ex vivo cellular cholesterol efflux was determined using TopFluor-cholesterol, a fluorescent cholesterol probe in which the cholesterol molecule is linked to a boron dipyrromethene difluoride (BODIPY) moiety (Avanti Polar Lipids, Alabaster, AL, USA) [21]. Briefly, 7.5 × 10^4^ J774A.1 cells/well were seeded in 48-well plates and allowed to grow for 24 h in a RPMI-supplemented medium. At that time point, the cells were labeled for 1 h in a low-glucose DMEM medium (Gibco, Waltham, MA, USA) containing 0.125 mmol/L total cholesterol, where the fluorescent cholesterol accounted for 20% of the total cholesterol complexed with 10 mmol/L methyl-β-cyclodextrin (Sigma-Aldrich, Madrid, Spain). The labeled cells were subsequently equilibrated for 18 h with DMEM containing 0.2% fatty-acid free BSA (Sigma-Aldrich) and then incubated for 24 h with 1% APOB-depleted serum from mice of each group in DMEM. The acyl-CoA cholesterol acyltransferase (ACAT) inhibitor Sandoz 58-035 (5 µmol/L; Sigma Aldrich, Madrid, Spain) was present during the whole experimental procedure. Fluorescence intensity was then measured in the medium using the microplate reader Synergy HT (BioTek Instruments, Winooski, VT, USA) at λEx/Em = 485/528 nm. The cells were solubilized with 1% cholic acid and mixed on a plate shaker for 1 h at room temperature, and fluorescence intensity was quantified. The cholesterol efflux capacity was calculated according to the formula: [media fluorescence/(media fluorescence + cells fluorescence)] × 100. All conditions were run in triplicate.

### 2.7. Bile Acid Analyses

The BA profiles of the mice given the ROO, FVOO, PE, or saline were determined from the feces collected for 48 h. Silylation of BA was carried out as previously reported, with some modifications [22]. Briefly, 200 µL of pyridine and 100 µL of *N*-methyl-*N*-(trimethylsilyl) rifluoroacetamide (Sigma-Aldrich) were added to vials containing 10 mg of vortexed feces, and then maintained at 60 °C for 30 min. After silylation, the samples were centrifuged at room temperature for 10 min at 8784× *g*. The supernatants were analyzed by GC-MS technology (GC-MS; Agilent 5973 MSD, Agilent Technologies, Madrid, Spain) following previously reported conditions [22]. Peak identification was based on a comparison between the retention times and mass fragmentation patterns and the reference compounds. For fecal BA quantification, commercial standards for cholic acid, deoxycholic acid, chenodeoxycholic acid, and lithocholic acid (Sigma-Aldrich, Madrid, Spain) were used. All quantifications were performed in the selected ion monitoring mode using calibration curves generated from different known concentrations of commercial standards.

### 2.8. Phenolic Biological Metabolites Determination

In the four groups of independent animals, serum samples from the mice were obtained after 2, 6, or 24 h post-intake (ROO, FVOO, PE, or saline) to determine the phenolic metabolites, as previously described [23]. In order to clean-up the biological matrix and preconcentrate the phenolic metabolites, the serum samples were pretreated by microelution extraction plates. Briefly, the cartridges were first conditioned sequentially using 250 μL of methanol and 250 μL of acidified Milli-Q water. Then, 350 μL of the serum sample mixed with 300 μL of 4% phosphoric acid and 50 μL of catechol (internal standard) at 10 mg/L was loaded onto the plate. A clean-up of the plates was performed within each sample. Finally, the retained phenolic metabolites were eluted with 2 × 50 μL of methanol. After extraction, the phenolic metabolites were determined by UPLC-MS). The chromatographic system consisted of an AcQuity™ UPLC equipped with a Waters binary pump system (Milford, MA, USA) using an AcQuity UPLC™ BEH C18 column (1.7 μm, 100 mm × 2.1 mm i.d.). UPLC-MS conditions were set up according to a previously reported protocol [23]. Phenolic metabolites were detected and quantified from their ion fragmentation in the MS/MS mode using the selected ion monitoring (SRM) mode, [23]. The SRM transitions, cone voltage, and collision energy values were optimized for each phenol metabolite to quantify the generated metabolites. The phenolic metabolites were quantified using their own calibration curve, performed in serum (the same biological matrix).

### 2.9. Cell Culture and Treatment

J774A.1 cells were seeded in 12-well plates at 5 × 10^5^ cells/well and grown until confluence was reached in the RPMI 1640-supplemented medium. Then, the cells were treated for 4 h with 0.06, 0.12, and 0.24 mg/mL PE (which corresponds to equivalent concentrations of 0.5, 1, and 2 µmol/L of hydroxytyrosol, respectively).

### 2.10. Quantitative RT-PCR Analyses

The total liver RNA was extracted using the TRIzol LS Reagent (Invitrogen, Carlsbad, CA, USA) following the manufacturer’s instructions and purified using the EZ-10 DNAaway RNA Miniprep Kit (Bio Basic, Markham, ON, Canada). The total RNA from the J774A.1 macrophages was isolated using the EZ-10 DNAaway RNA Miniprep Kit. cDNA was generated using Oligo(dT)_23_ and dNTPs mix (Sigma-Aldrich) and M-MLV reverse transcriptase RNase H minus point mutant (Promega, Madison, WI, USA). It was then subjected to quantitative Real-Time PCR amplification using the GoTaq(R) Probe qPCR Master Mix (Promega). Specific TaqMan probes (Applied Biosystems, Foster City, CA, USA) were used for *Apoa1* (Mm00437569_m1), *Abca1* (Mm004426464_m1), *Abcg1* (Mm00437390_m1), *Scarb1* (Mm00450236_m1), *Abcg5* (Mm00446241_m1), *Abcg8* (Mm00445970_m1), *Cyp7a1* (Mm00470430_m1), *Cyp27a1* (Mm00470430_m1), *Cyp7b1* (Mm00484157_m1), and *Rn18s* (Mm03928990_g1) was used as internal control gene. Reactions were run on a CFX96TM Real-Time System (Bio-Rad, Hercules, CA, USA) according to the manufacturer’s instructions. The relative mRNA expression levels were calculated using the ΔΔCt method.

### 2.11. Statistical Methods

Since the data followed a Gaussian distribution, a one-way ANOVA with a Tukey’s multiple comparisons post-test was used to compare the groups, while Dunnet’s post-test was used to compare the groups to a control group. The nonparametric Dunn’s post-test was used for data that did not follow the Gaussian distribution. GraphPad Prism 6.0 software (GraphPad, San Diego, CA, USA) was used to perform all statistical analyses. A *p* value < 0.05 was considered statistically significant.

## 3. Results

### 3.1. FVOO Increases HDL Cholesterol, APOA1, and the Formation of Nascent preβ-HDL Particles

Lipoprotein profiles and APOA1 levels were analyzed in female C57BL/6 mice administered ROO, FVOO, PE, or saline as a vehicle for 14 days. The mice tolerated the administration of the different supplements well, and no significant differences were observed in body weight between the groups (Table 1). At the end of the treatment, the FVOO and PE groups showed increased HDL-C levels compared to those of the saline and ROO groups (Table 1). FVOO intake also resulted in significant increases in APOA1 and preβ-HDL particles. However, none of these treatments had effects on activities of two major HDL remodeling proteins, i.e., LCAT and PLTP (Table 1).

### 3.2. FVOO and Its Phenolic Compounds Promote Macrophage-to-Feces Reverse Cholesterol Transport

To elucidate whether phenol-rich OO or its PC affected the whole-body macrophage-to-feces RevCT pathway in vivo, radiolabeled macrophages were injected into the mice at the end of treatment. [^3^H]tracer recovery was measured in serum at 4, 24, and 48 h post-injection, in the liver at 48 h, and in feces collected over 48 h. [^3^H]cholesterol levels were not affected by the treatment in either the serum or liver (Figure 1A,B). However, the radioactivity in the feces (cholesterol + BA) was significantly higher in both the FVOO- and PE-administered mice when compared to that of the saline and ROO groups (Figure 1C). Overall, this indicated that both FVOO and PE intake enhanced the transfer of macrophage-derived cholesterol to feces in vivo. The enhancement in the macrophage-specific RevCT rate was mainly due to a significant increase in the content of fecal [^3^H]BA in both the FVOO- and PE-administered mice when compared to that of the saline and ROO groups (0.10 ± 0.01% in saline, 0.10 ± 0.02% in ROO, 0.22 ± 0.03% in FVOO, and 0.18 ± 0.02% in PE). The [^3^H]cholesterol fraction was also increased in the FVOO and PE-administered mice; this change, however, was only significant when the PE and saline groups were compared (0.28 ± 0.02% in saline, 0.28 ± 0.04% in ROO, 0.35 ± 0.03% in FVOO, and 0.37 ± 0.03% in PE).

### 3.3. FVOO and OO Phenolic Compounds Upregulate HDL-Mediated Macrophage Cholesterol Efflux But Do Not Affect Liver Gene Expression Profile or Intestinal Cholesterol Absorption

To elucidate the potential of FVOO and the PE in promoting ex vivo HDL-mediated cholesterol efflux, the ability of the APOB-depleted post-treatment (ROO, FVOO, PE, or saline) serum was tested in TopFluor-cholesterol-labeled J774A.1 macrophages for a period of 4h. As shown in Figure 2A, FVOO intake increased the ability of the APOB-depleted serums to promote ex vivo cholesterol efflux when compared to the saline, ROO, and PE groups. PE APOB-depleted serum samples more greatly increased the macrophage cholesterol efflux when compared to the ROO, although this effect was less pronounced than that of the FVOO (Figure 2A). Macrophage cholesterol efflux capacity correlated with HDL-C levels when fitted to a linear model (*r* = 0.59, *p* < 0.0001). The liver expression of the genes involved in cholesterol trafficking was also evaluated. However, the FVOO and PE groups did not show significant alterations in their gene expression profiles, except for *Apoa1*, which was upregulated in the FVOO livers when compared to the PE, and *Cyp7a1*, which was downregulated in the PE livers when compared to the ROO livers (Figure 2B). The analyses of intestinal cholesterol absorption only revealed a moderate but significant reduction in the FVOO group when compared to the saline-treated mice but not when compared to the ROO or PE groups (Figure 2C). In addition, we determined the total bile acid profile in the mouse feces. We only found a significant increase in fecal chenodeoxycholic acid from the FVOO mice when compared with that of the saline group (Table 2).

### 3.4. OO Phenolic Compounds Upregulate Macrophage Abca1 Expression

We further evaluated serum phenol metabolite concentrations in the mice 2, 6, and 24 h after the administration of the ROO, FVOO, PE, or saline. Hydroxytyrosol sulfate was one of the main phenolic metabolites detected, and it was found to be significantly increased 2 h after the administration of FVOO and PE compared to the ROO and saline groups (Figure 3A). The hydroxytyrosol sulfate levels were also increased by both the FVOO and PE 6 h after their administration but to a lower extent in comparison to the levels found at 2 h. Serum hydroxytyrosol sulfate concentration returned to baseline levels after 24 h of administration and were similar in all groups (Figure 3A). Next, we analyzed the effects of the PE on macrophage transporters in vitro. For this purpose, J774A.1 macrophages were treated with 0.06, 0.12, and 0.24 mg/mL of PE (corresponding to equivalent concentrations of 0.5, 1, and 2 µmol/L of hydroxytyrosol present in the PE, respectively) for 4 h and the gene expression profile of the main receptors involved in the macrophage cholesterol efflux was analyzed. *Abca1* gene expression was upregulated by the PE, mainly at the dose of 0.12 mg/mL (Figure 3B). However, *Abcg1* and *Scarb1* expression was not modified by the PE.

## 4. Discussion

Phenol-rich OO intake may increase HDL-mediated cholesterol efflux capacity in both healthy and hypercholesterolemic individuals, although this effect has not been consistently found in all studies [7,11]. In the current study, we evaluated the effects of phenol-rich OO on whole-body RevCT from macrophages to feces in vivo for the first time. We used functional phenol-enriched FVOO (500 mg PC/kg of oil) with its own PC that had also been given to hypercholesterolemic subjects in the VOHF trial [7]. We found that FVOO enhances whole-body macrophage-to-feces RevCT in vivo in female C57BL/6 mice. One group of mice was administered saline as a control, and two additional groups of mice were administered ROO or PE in order to discern whether the effects of the FVOO were mainly due to the OO fat or the OO PC. PE also enhanced the RevCT rate, whereas ROO devoid of PC did not. These results are in line with the reported absence of effects of a dietary fat source (as saturated or polyunsaturated fatty acids) on in vivo macrophage-specific RevCT in mice [24,25]. Overall, our data strongly indicates that the amount of dietary fat appears to have minor effects on the RevCT rate in vivo. Rather than monounsaturated fatty acids (MUFAs), the PCs presented in the VOO are the main inducers of the macrophage-specific RevCT rate.

Two recent meta-analyses have found that phenol-rich OO intake increases HDL-C [26,27], mostly in healthy subjects [27]. In particular, phenol-rich OO intake increased HDL-C in 200 healthy volunteers from the EUROLIVE study [6], although these changes have not been observed in a subsample of the whole EUROLIVE study population [7]. HDL-C levels also remain unchanged when hypercholesterolemic volunteers ingest FVOO [11]. In our study, FVOO administered mice also showed increased levels of HDL-C, nascent preβ-HDL particles, and APOA1. Nevertheless, the HDL-C increase was more limited in the PE-administered mice, and no changes were observed in their serum APOA1 levels, indicating that PCs *per se* were not responsible for all these changes.

The RevCT is a multistep process that begins with cholesterol efflux from peripheral cells. We tested the cholesterol efflux capacity of APOB-depleted serums (which contain HDL and all regulatory HDL components, including lipid transfer proteins and enzymes) from mice administered with saline, ROO, FVOO, or PE. Both FVOO and PE administration increased the ex vivo cholesterol efflux from the macrophages. Of note, cholesterol efflux tended to be higher in the mice given the FVOO, which could be related to the differential PC patterns of FVOO and PE; however, these relatively small changes did not result in differential effects on the macrophage RevCT rate. These findings are also in line with the results from the EUROLIVE study, in which phenol-rich OO intake increased the HDL-mediated cholesterol efflux capacity [7]. However, the FVOO and PE did not modify the macrophage-derived radioactivity in either the serum or liver. It should be noted that RevCT is a dynamic process, and [^3^H]cholesterol levels in intermediate compartments at any time point do not always reflect the continuous flux through different body compartments. Indeed, the amount of [^3^H]tracer in the feces collected over 48 h, the ultimate destiny of macrophage-derived cholesterol, gives a cumulative quantity of the cholesterol released by the macrophages and transported to the feces [13]. Moreover, the expression of the genes involved in liver cholesterol trafficking was not consistently modified by the FVOO and PE treatments. We also determined whether OO PCs modify intestinal cholesterol absorption, as interventions that inhibit cholesterol absorption usually enhance the rate of the macrophage-to-feces RevCT pathway [12]. FVOO administration caused a moderate but significant decrease in intestinal cholesterol absorption. However, this result could not fully explain the increase in the macrophage-to-feces RevCT induced by the OO PCs, as both the FVOO and PE induced a ≈50% increase in the macrophage-specific RevCT. In contrast, only the FVOO showed a ≈5.5% reduction in intestinal cholesterol absorption.

The induction in the macrophage-derived fecal [^3^H]tracer by FVOO and PE was mainly due to an increase in [^3^H]BA excretion. Early reports have indicated the potential of dietary MUFA on bile secretion [28,29], although this change appeared to be disassociated with the expression of the main liver BA biosynthetic rate-limiting enzyme, *Cyp7a1* [28]. FVOO administration neither changed the expression of *Cyp7a1* nor the expression of the main enzymes involved in the alternative BA biosynthetic pathway (*Cyp27a1* and *Cyp7b1*). ROO tended to upregulate *Cyp7a1*, whereas the PE downregulated its expression. Therefore, the MUFA content of OO appears to induce the expression of *Cyp7a1*, counteracted by the PC present in the FVOO. We found a moderate but significant increase in the fecal excretion of primary BA chenodeoxycholic acid after FVOO consumption (13%). This was completely not associated with the expression of the liver BA biosynthetic genes. Since more than 90% of BA is reabsorbed in the ileum by active transport [30], the effects of FVOO on intestinal BA transporters deserve further investigation. Overall, our findings indicate that the increase in fecal [^3^H]BA is largely due to an enhanced macrophage-derived cholesterol transfer to the liver, which is converted to BA and released to the intestine. This observation coincides with the enhanced excretion of fecal macrophage-derived [^3^H]cholesterol.

Hydroxytyrosol is one of the most biologically active PCs present in VOO. We did not identify it in its free form, which is consistent with indications that this compound is rapidly subjected to phase II metabolism [31]. The main circulating hydroxytyrosol metabolite is hydroxytyrosol sulfate derivative [15,31], which we found increased in the serum after FVOO and PE administration. We next analyzed the effects of the PE on the macrophage cholesterol transporters. The PE upregulated *Abca1* gene expression in macrophages. This result is consistent with a previous report where the PC from extra VOO, including hydroxytyrosol, stimulates ABCA1 protein expression and in vitro APOA1-mediated cholesterol efflux in J774 macrophages [32,33]. PE-mediated effects on macrophage *Abca1* gene expression were observed to be relevant even at lower concentrations than those of hydroxytyrosol sulfate in the serum. It should be noted that the peritoneal cavity and intimal fluids are filtrates from blood plasma [34], and, thus, the PC metabolites concentrations that arrive at the macrophages are likely to be lower than those from the circulation.

A limitation of this study is that both FVOO and PE were tested in female C57BL/6 mice and we cannot rule that these compounds could affect differentially macrophage RevCT rate in male mice, which are highly susceptible to develop obesity and insulin resistance. The potential of these compounds to modulate RevCT in mouse models of hyperlipidemia and atherosclerosis also deserve further investigation.

## 5. Conclusions

Our study demonstrates that the OO PCs promotes macrophage-specific RevCT in vivo in female C57BL/6 mice. This change was closely related to the ability of the PE to induce macrophage *Abca1* expression, thereby increasing its potential to facilitate HDL-mediated cholesterol efflux. The enrichment of VOO with the PCs could be a way of increasing the beneficial properties of VOO without raising its caloric content, constituting a nutraceutical strategy to enhance HDL cardioprotective properties.

## Figures and Tables

**Figure 1 biomedicines-08-00266-f001:**
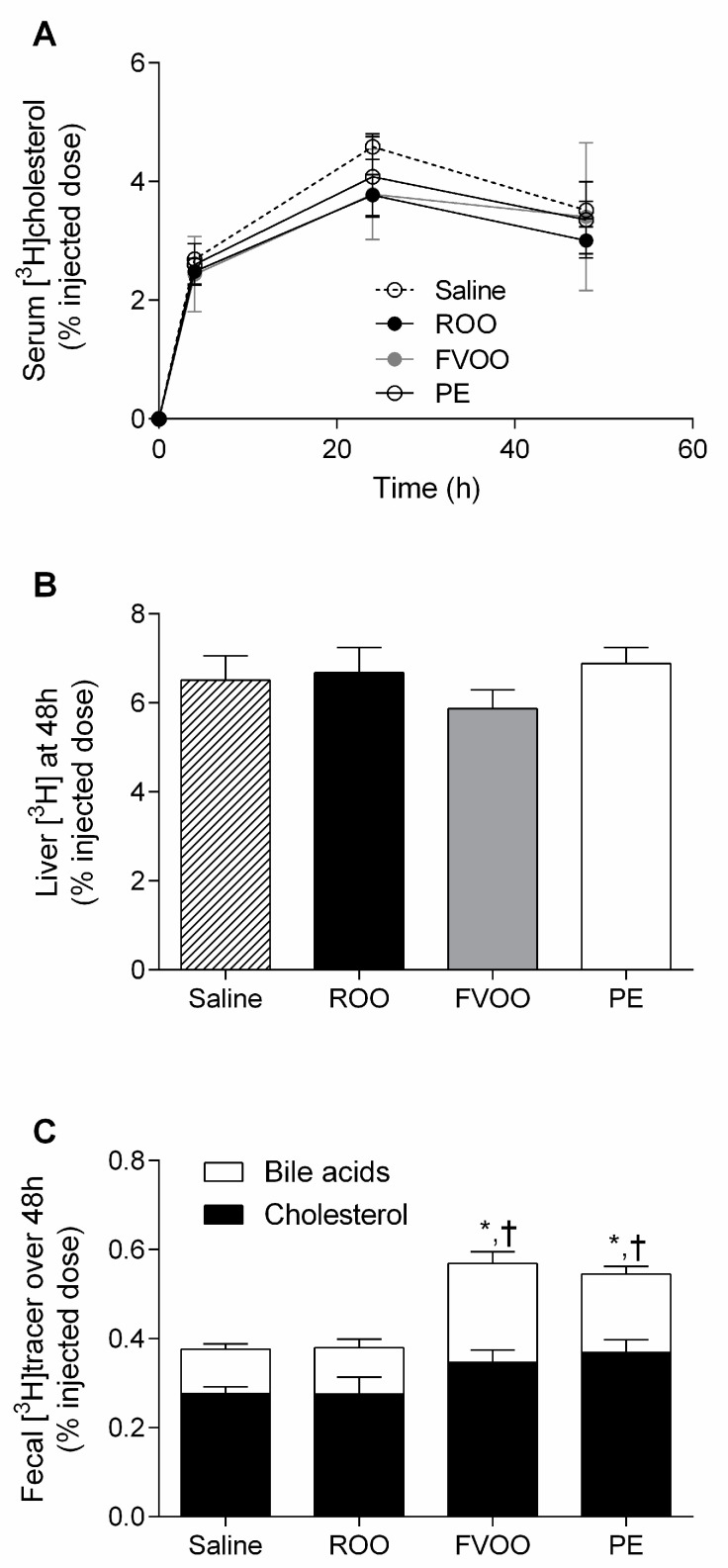
FVOO and its phenolic compounds promote macrophage-to-feces RevCT. (**A**) C57BL/6 mice were administered intragastric doses of ROO, FVOO, PE, or the vehicle (saline) for 14 days. Individually-housed mice were intraperitoneally injected with [^3^H]cholesterol-labeled J774A.1 macrophages, and serum [^3^H]cholesterol was determined at the indicated times. (**B**) Liver [^3^H]cholesterol was determined at 48 h. (**C**) [^3^H]cholesterol and [^3^H]tracer from fecal bile acids collected over 48 h. Values are the mean ± SEM of 10 mice in both saline and PE groups and 8 mice in both ROO and FVOO groups. * *p* < 0.05 vs. saline and ^†^
*p* < 0.05 vs. ROO.

**Figure 2 biomedicines-08-00266-f002:**
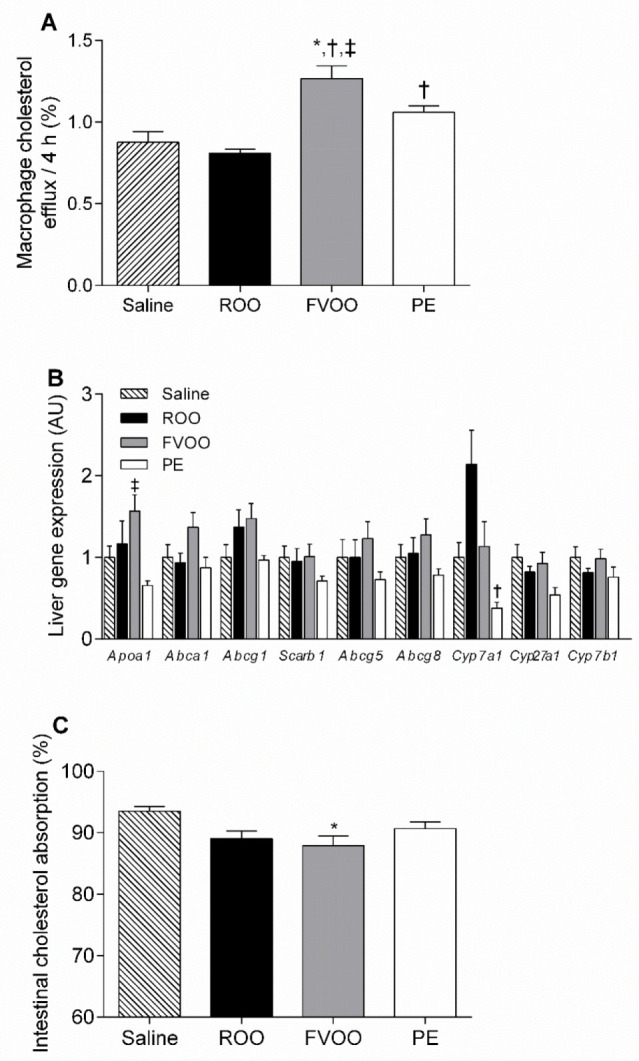
FVOO and its phenolic compounds upregulate macrophage cholesterol efflux but did not affect liver gene expression profile or intestinal cholesterol absorption. (**A**) TopFluor-cholesterol-labeled J774A.1 macrophages were incubated with a media containing 1% APOB-depleted serum from C57BL/6 mice given intragastric doses of ROO, FVOO, PE, or the vehicle (saline) for 24 h, and cholesterol efflux to media was determined. Values presented as mean ± SEM of 10 mice in saline group, 9 mice in ROO group, 11 mice in FVOO group, and 15 mice in PE group. (**B**) Relative liver gene expression profile. The signal of the saline group was set to a normalized value of 1 arbitrary unit (AU) for each gene. Results are presented as mean ± SEM of 9 mice in saline group, 5 mice in both ROO and PE groups, and 10 mice in FVOO group. (**C**) Fractional intestinal cholesterol absorption was determined by a fecal dual-isotope ratio method in 8 mice in saline group and 9 mice in ROO, FVOO, and PE groups. * *p* < 0.05 vs. saline, ^†^
*p* < 0.05 vs. ROO, and ^‡^
*p* < 0.05 vs. PE.

**Figure 3 biomedicines-08-00266-f003:**
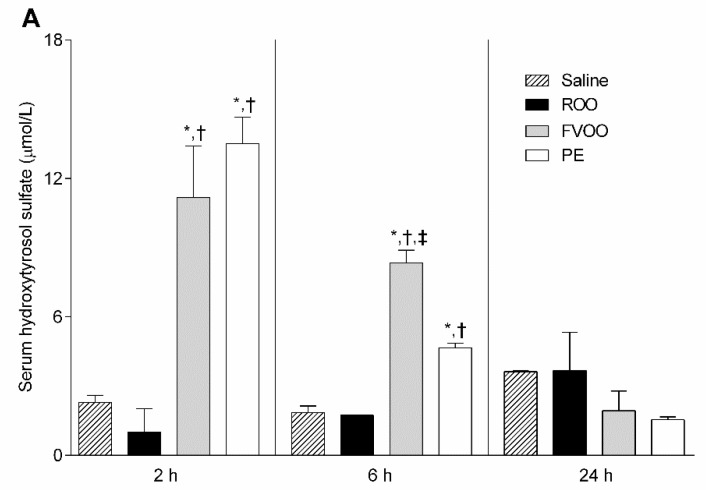
OO phenolic compounds upregulate macrophage *Abca1* expression. (**A**) Hydroxytyrosol sulfate levels in the serum after 2, 6, and 24 h post-administration of ROO, FVOO, PE, or the vehicle (saline). Values are the mean ± SEM of 3 individual mice/group. * *p* < 0.05 vs. saline, ^†^
*p* < 0.05 vs. ROO, and ^‡^
*p* < 0.05 vs. PE, analyzed by ANOVA Tuckey’s multiple comparison test. (**B**) Real-time PCR quantification of relative mRNA expression in J774A.1 macrophages treated with PE (0.06, 0.12, and 0.24 mg/mL of PE correspond to 0.5, 1, and 2 µM of hydroxytyrosol present in the PE, respectively) for 4 h. The signal of control J774A.1 macrophages without the PE was set to a normalized value of 1 AU for each gene. Three independent experiments were performed for each condition. * *p* < 0.05 vs. 0 mg/mL PE.

**Table 1 biomedicines-08-00266-t001:** Serum and HDL parameters in female C57BL/6 mice administered saline, ROO, FVOO or PE for 14 days.

	Saline n = 20	ROO n = 18	FVOO n = 19	PE n = 14
Body weight (g)	20.00 ± 0.55	20.16 ± 0.45	20.73 ± 0.70	20.36 ± 0.63
Total cholesterol (mmol/L)	2.28 ± 0.08	2.28 ± 0.10	2.86 ± 0.15 *^,†^	2.74 ± 0.12 *
VLDL+LDL cholesterol (mmol/L)	0.56 ± 0.03	0.50 ± 0.04	0.57 ± 0.04	0.60 ± 0.04
HDL cholesterol (mmol/L)	1.72 ± 0.08	1.79 ± 0.10	2.29 ± 0.13 *^,†^	2.15 ± 0.10 *
APOA1 (g/L)	1.13 ± 0.04	1.22 ± 0.01	1.38 ± 0.06 *	1.20 ± 0.05
Preβ-HDL particles (g/L)	0.13 ± 0.01	0.14 ± 0.01	0.18 ± 0.02 *^,‡^	0.13 ± 0.01
PLTP activity (nmol/mL/h)	15,725 ± 1169	17,918 ± 1099	16,920 ± 1008	18,336 ± 908
LCAT activity (nmol/mL/h)	18.39 ± 1.28	17.55 ± 1.52	18.02 ± 0.44	19.70 ± 1.03

ROO: refined olive oil, FVOO: functional virgin olive oil, PE: phenolic extract, VLDL: very-low-density lipoprotein, LDL: low-density lipoprotein, HDL: high-density lipoprotein, APOA1: apolipoprotein A1, PLTP: phospholipid transfer protein, LCAT: lecithin–cholesterol acyltransferase. Results are presented as mean ± SEM. * *p* < 0.05 vs. saline, ^†^
*p* < 0.05 vs. ROO, and ^‡^
*p* < 0.05 vs. PE.

**Table 2 biomedicines-08-00266-t002:** Fecal bile acids in C57BL/6 mice administered saline, ROO, FVOO, or PE for 14 days.

	Saline	ROO	FVOO	PE
Cholic acid (µg/g feces)	14.67 ± 1.06	12.38 ± 1.14	10.76 ± 0.67	12.42 ± 0.97
Chenodeoxycholic acid (µg/g feces)	7.68 ± 0.08	8.32 ± 0.34	8.71 ± 0.06 *	8.27 ± 0.29
Deoxycholic acid (µg/g feces)	22.78 ± 0.49	20.51 ± 4.06	23.43 ± 2.18	20.21 ± 1.16
Lithocholic acid (µg/g feces)	5.76 ± 0.22	6.65 ± 0.63	7.58 ± 0.40	6.42 ± 0.58

Results are presented as mean ± SEM of 5 mice/group. * *p* < 0.05 vs. saline.

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
