# Peer review of "Phenol-Enriched Virgin Olive Oil Promotes Macrophage-Specific Reverse Cholesterol Transport In Vivo"

_biomedicines, 2020, doi:10.3390/biomedicines8080266_

Round 1
Reviewer 1 Report
Reviewer Evaluation:
This research paper assesses the effect of unrefined phenol-enriched virgin olive oil on the macrophage-induced reverse cholesterol transport in mice. This study uses several in vitro, in vivo, and ex vivo strategies to determine the mechanisms involved. While I enjoyed reading the study, there are some outstanding issues that require attention. I look forward to the authors responses and good luck.
Major points, improvement or queries:
- For clarity, I suggest a title change as follows “Unrefined phenol-enriched virgin olive oil promotes macrophage-specific reverse cholesterol transport in vivo” or something to that effect if the authors deem it appropriate.
- Why was saline chosen as a vehicle? I assume the olive oil did not disperse evenly in saline due to solubility issues. Would BSA-saline not have been more appropriate as the lipids disperse better in BSA? If the oil floated to the top and was not evenly dispersed how could you have ensured the same ratio of vehicle:oil every time. The oil would be different every time. Are there other vehicles that might have been more appropriate? You say 150µL of the oils were given via intragastric doses, how much saline accompanied those doses, was it an equal proportion? Thank you.
- Optional edit: As RCT is a well-recognised abbreviation for randomised control trial, I would suggest changing your abbreviation for reverse cholesterol transport to ‘RevCT’ or something to that effect.
- Line 69/70 Olive has been synonymous with health benefits. The idea that you would have to fortify olive oils with its own constituents further seems like a redundant exercise? It is notable that he lipids within olive oil are associated with anti-atherogenic compounds such as polar lipids or glycolipids that reduce foam cell formation. By adding extra of the PC to the olive oil, the ratio of the other beneficial compounds would reduce. I’d love to hear the authors thoughts on almost refortifying olive oil with its own constituents.
- Section 2.2/2.4 – briefly mention the plasma preparation in a sentence so that others can see your methods.
- Line 137 – is 48 hrs long enough to induce an effect?
- Why were only female mice studied? Was sexual dimorphism not considered? In humans, cardiovascular diseases differentially affect males and females. Likewise, olive oil contains a number of phytosterols that behave similarly to oestrogens etc. that may/may not affect male mice. Would alternative results have occurred if males were also investigated. Can the authors explain this. Maybe consider including this in a limitations paragraph in the discussion
- How many mice were given each treatment? Please report the N. Without that it is difficult to determine the meaning and significance of the outcomes.
- Saline seemed to do better for a number of parameters in Figure 2. It seems unlikely that salty water would have a more beneficial effect than a refined olive, which still has low levels, but still beneficial microconstituents. What do the authors think is going on?
- The article would benefit from a limitations section at the end of the discussion that also highlights future perspectives in this field.
Minor Comments:
Line 39 – replace ‘produced a significant’ with ‘led to a significant’
Line 37, 43, 175, 189, 233, 236, table 1 legend and footer, and throughout the discussion – saline should on have a small ‘s’
Line 68 – ‘..concentration is much too low in commercial virgin OO (VOO) to reach…’
Line 93 – write ‘twelve’ for consistency
Line 95 – was this chow? Add a little more detail on the food (manufacturer/source).
Line 169 – microplate reader (no caps)
Line 181 – delete extra punctuation point
Author Response
Reviewer 1 Evaluation:
This research paper assesses the effect of unrefined phenol-enriched virgin olive oil on the macrophage-induced reverse cholesterol transport in mice. This study uses several in vitro, in vivo, and ex vivo strategies to determine the mechanisms involved. While I enjoyed reading the study, there are some outstanding issues that require attention. I look forward to the authors responses and good luck.
We acknowledge the reviewer for her/his comments and suggestions which have helped us to significantly improve the manuscript. Our responses are presented below, and the changes have been highlighted in the revised version of the manuscript.
Major points, improvement or queries:
- For clarity, I suggest a title change as follows “Unrefined phenol-enriched virgin olive oil promotes macrophage-specific reverse cholesterol transport in vivo” or something to that effect if the authors deem it appropriate.
Thank you very much for the suggestion, the title has been modified to better explain the focus of the study: Phenol-enriched virgin olive oil promotes macrophage-specific reverse cholesterol transport in vivo”. To clarify the olive oils employed in the present manuscript we have added the following explanation in the sentence: ”In the present study, we determined the potential of a phenol-enriched VOO (thus unrefined as required by the virgin OO EU classification) and its own PC to promote macrophage-to-feces RevCT in vivo.”
In this regard, we would like to highlight that virgin and extra-virgin olive oil are obtained by direct first-pressing or centrifugation of olives. The criteria to certify an olive oil as extra-virgin are: 1/ no more than 0.8% of oleic acidity; 2/ mechanical extraction methods without chemicals and hot water; 3/ first cold-pressing; and 4/ perfect taste. Virgin olive oils have a maximum free acidity of 2%, those with an acidity greater than 3.3% are submitted to a refining process in which mainly phenolic compounds and, to a lesser extent, squalene are lost. After refination the name of virgin olive oil cannot be further applied, in this regard ordinary olive oil is produced by mixing virgin and refined olive oil [EC. Regulation 1513. Off J Eur Com, L 201, 2001 [26.07.2003].
- Why was saline chosen as a vehicle? I assume the olive oil did not disperse evenly in saline due to solubility issues. Would BSA-saline not have been more appropriate as the lipids disperse better in BSA? If the oil floated to the top and was not evenly dispersed how could you have ensured the same ratio of vehicle:oil every time. The oil would be different every time. Are there other vehicles that might have been more appropriate? You say 150µL of the oils were given via intragastric doses, how much saline accompanied those doses, was it an equal proportion? Thank you.
In this study, we administered daily intragastric doses of refined olive oil (ROO) and OO enriched with its own PC (FVOO, 500 mg PCs/kg of oil) respectively, for 14 days. Saline was not administered in these two groups. In parallel, we administered the phenol extract (PE) used in FVOO preparation and dissolved in a saline solution, or saline (the vehicle solution as control of PE), respectively, for 14 days. 150 µL/mouse/day were administered in all cases. These points have been clarified in Methods, section 2.1.
- Optional edit: As RCT is a well-recognised abbreviation for randomised control trial, I would suggest changing your abbreviation for reverse cholesterol transport to ‘RevCT’ or something to that effect.
As suggested by the reviewer, we have replaced RCT by RevCT through all the manuscript.
- Line 69/70 Olive has been synonymous with health benefits. The idea that you would have to fortify olive oils with its own constituents further seems like a redundant exercise? It is notable that he lipids within olive oil are associated with anti-atherogenic compounds such as polar lipids or glycolipids that reduce foam cell formation. By adding extra of the PC to the olive oil, the ratio of the other beneficial compounds would reduce. I’d love to hear the authors thoughts on almost refortifying olive oil with its own constituents.
Our preliminary data indicated that both virgin olive oil and OO enriched with its own PC (FVOO) promoted macrophage RevCT, but these effects were more pronounced after administering FVOO. Since we also found that PE dissolved in saline (used in the preparation of FVOO) promoted RevCT by itself at similar levels, we investigated the potential mechanisms underlying these HDL-mediated effects in both FVOO and PE groups (and ROO and saline as control groups, respectively). Indeed, our data indicate that OO PCs, rather than other OO compounds, are the major inductors of macrophage-specific RevCT in vivo.
- Section 2.2/2.4 – briefly mention the plasma preparation in a sentence so that others can see your methods.
As requested, we have specified how serum samples were obtained.
- Line 137 – is 48 hrs long enough to induce an effect?
As commented above, the animals were treated for 14 consecutive days, but the changes in accumulative macrophage RevCT rates were measured for 48h. In this assay, mouse macrophages were loaded with acetylated LDL and [3H]cholesterol and then injected intraperitoneally into recipient animals. After injection, the appearance of tracer in serum was determined at different time points and, most importantly, collected continuously in the feces during 48h. This method has been used to study the role of different therapies and pathways relevant for RevCT and HDL-mediated atheroprotection (reviewed in ref 13, Lee-Rueckert et al, BBA 2016). This point has been included at the end of the section 2.4.
- Why were only female mice studied? Was sexual dimorphism not considered? In humans, cardiovascular diseases differentially affect males and females. Likewise, olive oil contains a number of phytosterols that behave similarly to oestrogens etc. that may/may not affect male mice. Would alternative results have occurred if males were also investigated. Can the authors explain this. Maybe consider including this in a limitations paragraph in the discussion
Based in our previous data and our previous reports, macrophage RevCT rates are similar in both C57BL/6 mouse sexes (Atherosclerosis 196 (2008) 505–513). Given that male mice are more susceptible to obesity and insulin resistance, we conducted all mechanistic experiments in females to avoid the side effects of OO in these parameters and, their indirect effects on RevCT. We cannot rule the side effects of other minor FVOO compounds in female mice, although the animals administered with only PE showed the direct effects of OO PCs. As requested, we have included these points in a limitations paragraph.
- How many mice were given each treatment? Please report the N. Without that it is difficult to determine the meaning and significance of the outcomes.
The specific N in each group has been specified in each Table/Figure legends.
- Saline seemed to do better for a number of parameters in Figure 2. It seems unlikely that salty water would have a more beneficial effect than a refined olive, which still has low levels, but still beneficial microconstituents. What do the authors think is going on?
Indeed, ROO and saline did not affect macrophage cholesterol efflux (panel A), but ROO tended to upregulate liver abcg1 and cyp7a1 expression (panel B), which could be considered a positive effect although, unfortunately, these changes were not significantly altered and appeared not to be enough for regulating bile acid excretion (Table 2). Furthermore, ROO also tended to reduce intestinal cholesterol absorption (panel C), but, again, these changes were not significant.
- The article would benefit from a limitations section at the end of the discussion that also highlights future perspectives in this field.
As requested, we have added a limitations paragraph including that the potential of both FVOO and PE to modulate RevRCT in mouse models of hyperlipidemia and atherosclerosis deserve further investigation.
Minor Comments:
Line 39 – replace ‘produced a significant’ with ‘led to a significant’
We have replaced it.
Line 37, 43, 175, 189, 233, 236, table 1 legend and footer, and throughout the discussion – saline should on have a small ‘s’
We have included it.
Line 68 – ‘concentration is much too low in commercial virgin OO (VOO) to reach…’ We have modified this sentence.
Line 93 – write ‘twelve’ for consistency
We have included it.
Line 95 – was this chow? Add a little more detail on the food (manufacturer/source). We have included it.
Line 169 – microplate reader (no caps)
We have replaced it.
Line 181 – delete extra punctuation point
Done.
Reviewer 2 Report
In this paper the efficacy of phenol enriched olive oil to promote reverse cholesterol transport from macrophage to feces has been proved in a murine model.
Overall, the topic is interesting, the work brings many experiments, and is well supported by the literature. A criticism concerns the abstract. It must be rewritten because it is confusing and does not reflect the main text. I also suggest changing the title, and deleting "but not refined" because there is no mention of this in the text.
Author Response
Reviewer 2 Evaluation:
In this paper the efficacy of phenol enriched olive oil to promote reverse cholesterol transport from macrophage to feces has been proved in a murine model.
Overall, the topic is interesting, the work brings many experiments, and is well supported by the literature. A criticism concerns the abstract. It must be rewritten because it is confusing and does not reflect the main text. I also suggest changing the title, and deleting "but not refined" because there is no mention of this in the text.
We thank the reviewer for her/his comments and suggestions, which have helped us improve the manuscript. We have changed the title by “Phenol-enriched virgin olive oil promotes macrophage-specific reverse cholesterol transport in vivo. ” To clarify the olive oils employed in the present manuscript we have added the following explanation in the sentence: ”In the present study, we determined the potential of a phenol-enriched VOO (thus unrefined as required by the virgin OO EU classification) and its own PC to promote macrophage-to-feces RevCT in vivo.”
As requested, we have also rewritten the abstract to include our main findings, but limited to 200 words.
Round 2
Reviewer 1 Report
I would like to thank the authors for clarifying my questions, one of which I realise I misunderstood so I appreciate the explanation. Congratulations on a very interesting study.